# Advances in the Application of Small Unoccupied Aircraft Systems (sUAS) for High-Throughput Plant Phenotyping

**Ibukun T. Ayankojo [1], Kelly R. Thorp [2] and Alison L. Thompson [3],***

1   Department of Plant and Soil Sciences, North Mississippi Research and Extension Center, Verona, MS 38879, USA
2   USDA-ARS, Arid Land Agricultural Research Center, Maricopa, AZ 85138, USA
3   USDA-ARS, Wheat Health, Genetics, and Quality Research Unit, Pullman, WA 99164, USA
*   Correspondence: alison.thompson@usda.gov

**Abstract:** High-throughput plant phenotyping (HTPP) involves the application of modern information technologies to evaluate the effects of genetics, environment, and management on the expression of plant traits in plant breeding programs. In recent years, HTPP has been advanced via sensors mounted on terrestrial vehicles and small unoccupied aircraft systems (sUAS) to estimate plant phenotypes in several crops. Previous reviews have summarized these recent advances, but the accuracy of estimation across traits, platforms, crops, and sensors has not been fully established. Therefore, the objectives of this review were to (1) identify the advantages and limitations of terrestrial and sUAS platforms for HTPP, (2) summarize the different imaging techniques and image processing methods used for HTPP, (3) describe individual plant traits that have been quantified using sUAS, (4) summarize the different imaging techniques and image processing methods used for HTPP, and (5) compare the accuracy of estimation among traits, platforms, crops, and sensors. A literature survey was conducted using the Web of Science™ Core Collection Database (THOMSON REUTERS™) to retrieve articles focused on HTPP research. A total of 205 articles were obtained and reviewed using the Google search engine. Based on the information gathered from the literature, in terms of flexibility and ease of operation, sUAS technology is a more practical and cost-effective solution for rapid HTPP at field scale level (>2 ha) compared to terrestrial platforms. Of all the various plant traits or phenotypes, plant growth traits (height, LAI, canopy cover, etc.) were studied most often, while RGB and multispectral sensors were most often deployed aboard sUAS in HTPP research. Sensor performance for estimating crop traits tended to vary according to the chosen platform and crop trait of interest. Regardless of sensor type, the prediction accuracies for crop trait extraction (across multiple crops) were similar for both sUAS and terrestrial platforms; however, yield prediction from sUAS platforms was more accurate compared to terrestrial phenotyping platforms. This review presents a useful guide for researchers in the HTPP community on appropriately matching their traits of interest with the most suitable sensor and platform.

**Keywords:** high-throughput plant phenotyping; crop traits; platform comparison

## 1. Introduction

Global agriculture faces several challenges, including the growing human population, resource and environmental sustainability, novel crop pests and diseases, and climate change [1–3]. For example, stresses related to water deficit during crop production are a leading cause of production loss on a global scale [4]. The increase in air temperatures under both current and future climatic conditions will increase evapotranspiration, which will increase crop water demand [5]. This condition not only reduces crop productivity [5,6] but also increases water withdrawal for agricultural purposes, resulting in greater pressure on already limited water resources [5–9]. Significant reductions in crop growth and productivity have also been reported under other stress conditions such as heat and increased soil salinity [10–14]. To overcome these challenges, understanding genotype and

phenotype interactions are necessary to facilitate plant breeding for heat, drought, and disease-resilient crop cultivars that are high yielding and resource-use efficient [15]. This is because enhanced crop resilience to stress conditions has been reported to improve crop growth with minimal or no impacts on productivity [16–19].

Over the last two decades, technologies for crop improvement, such as whole-genome sequencing, have developed rapidly [15]. However, due to the limitations of phenotyping methods, knowledge remains limited on the linkages between genetic information and the phenotypic traits associated with crop growth, yield, and stress adaptation [20–22]. The conventional approach of phenotyping-by-eye, which has been the mainstay of selection breeding for more than 10 decades, does not provide the required throughput to effectively link phenotypes with the vast genetic information available through genome sequencing [23]. Crop phenotyping using this conventional method is slow and labor-intensive, hence not time-efficient for a large-scale breeding program; it is also not cost-efficient, and the data are often influenced by human error [24]. New technologies for measuring phenotypic traits at large scales and throughout the growing season are needed to improve genomic selection models for breeding efficiency [25]. To fully harness the benefit of modern genomic information and associate that information with important crop phenotypes, the development of more efficient, reliable, and multifunctional High-Throughput Plant Phenotyping (HTPP) technologies is essential [15,20]. A primary goal of such HTPP technologies is to enable rapid and non-destructive large-scale estimation of plant morphological and physiological traits using image-based data processing pipelines [26,27].

Early works in HTPP were largely conducted on the model plant *Arabidopsis thaliana* [28] but has quickly been adopted by researchers in several important agricultural crops, including wheat [29], millet [30], rice [31], cotton [32], barley [33], maize [34], and sorghum [35]. Many of these early studies were conducted in controlled environments, but plant responses under controlled conditions are often not representative of responses under field conditions [29,36–38]. Moving the phenotyping efforts to the field sparked the development of terrestrial sensing platforms for field-based HTPP [20]. While these studies made substantial advances toward practical field-based HTPP technologies, the scalability issue of terrestrial HTPP platforms was noted, meaning their application may be limited when applied to large-scale phenotyping operations [25]. Depending on the vehicle design, the operational scale of terrestrial HTPP platforms is often limited due to sensor height, operational speed, and equipment maneuverability in the field. Previous studies have indicated that under field conditions, pole or tower-based HTPP platforms are functional only up to 50 to 120 m away from the target [39,40]. Additionally, the operational speed of terrestrial mobile carts is generally within 1 m s$^{-1}$ or slower due to the typical "stop-measure-go" data collection approach [24,41,42]. Unlike mobile carts, the gantry-type platforms are typically designed for continuous phenotyping without stopping [43,44]; however, the measurement area is limited by the coverage of the fixed platform. Therefore, the operational scale of terrestrial HTPP platforms is generally restricted to an individual plant, parcel, or plot scales (<2 ha) [42].

To overcome these limitations, small Unoccupied Aircraft Systems (sUAS) have been identified as a promising solution for rapid HTPP at larger field scales (>2 ha) [45]. The recent advances in remote-sensing technologies in combination with automatic control systems, aeronautics, and high-performance computing have enhanced the application of sUAS for HTPP. For example, sUAS has been successfully used for HTPP with a total coverage area ranging from 8 to 757 ha [46,47], owing to the recent advancement in hardware optimization and battery technology [48]. Therefore, despite their susceptibility to weather, payload capacity, endurance, and aviation regulation constraints, the benefits of sUAS platforms, including ease of scalability, flexibility in flight planning, and relatively low cost, have begun to shift plant phenotyping efforts from the ground to the air [48].

Common sUAS applications include estimating crop growth (LAI, plant height, canopy cover, etc.), identifying tolerance or adaptation to biotic (pest and diseases) and abiotic (temperature, salinity, drought, etc.) stressors, predicting crop yield, and determining

resource-use efficiencies (nutrients, water, radiation, etc.) [4,49–56]. Deployments of sUAS are not limited by the ground conditions (i.e., trafficability issues due to soil wetness, elevation gradient, or advanced crop growth stage), and sUAS can permit the rapid acquisition of high-resolution images over relatively large (<2 ha) field areas compared to terrestrial systems. The sUAS also permit frequent field observation and crop phenotyping throughout the crop growth cycle [53,57]. Due to the aforementioned advantages compared to terrestrial systems, sUAS has been recognized as a more practical and cost-effective platform for collecting crop images [15]. Although the adoption of sUAS-based HTPP platforms is increasing, they may not be suitable for all HTPP applications and thus may not be considered a total replacement for terrestrial platforms. For example, because image resolution decreases with increasing altitude, evaluation of crop traits that require detailed imaging (such as flower and fruit counts) may be less reliable using sUAS platforms. Additionally, compared to terrestrial platforms, sUAS operations and image quality may be limited by weather conditions (high wind) and payload capacity [24,58,59]. Due to the contrasting capabilities and practicalities of sUAS and terrestrial phenotyping platforms, further research is needed to identify the advantages and limitations of both strategies for various applications in HTPP.

To advance knowledge on the use and elucidate the appropriateness of sUAS for HTPP, the goals of this review were to describe recent advances in the application of sUAS technology for field-based HTPP by (1) identifying the advantages and limitations of terrestrial and sUAS platforms for HTPP (Section 1), (2) summarizing the different imaging techniques and image processing methods used for HTPP (Sections 2 and 3), (3) describing individual plant traits that have been quantified using sUAS (Section 4), and (4) comparing the accuracy of estimation among traits, platforms, crops, and sensors (Section 5). To date, several review papers have been conducted on the application of remote sensing platforms, including sUAS for crop data collection [15,20,45]. In 2013, Atzberger [60] summarized the advances in remote sensing in terms of regional and global applications for agriculture but not specifically for HTPP. Hassler and Baysal-Gure [61] conducted a summary of the application of sUAS technology in different agricultural systems. However, this review focused on sUAS applications for precision agriculture and not for crop improvement or breeding purposes. Lu et al. [1] followed a similar approach with a limited scope on hyperspectral sensors. While other reviews by Rebetzke et al. [23], Yang et al. [15], Li et al. [20], Araus and Cairns [62], and Gupta et al. [63] were focused on HTPP, these reviews were either focused on controlled environments and/or on general descriptions of trait estimation from previous research works. This present review expands on reviews conducted by Xie and Yang [45] and Feng et al. [25] while providing a new analysis. These reviews focused on detailed descriptions of the application of sUAS for HTPP but provided limited or no information on the accuracy of trait estimation between terrestrial and sUAS platforms. In addition to providing a comprehensive review of more recent studies in HTPP research, this present review classifies the accuracy of estimation from these studies based on traits, platforms, crops, and sensors. Therefore, this review expands the previous efforts by providing unique information to the research community on appropriately matching their crop traits of interest with a suitable sensor and platform.

*Literature Survey*

A literature survey was conducted using the Web of Science<sup>TM</sup> Core Collection Database (THOMSON REUTERS<sup>TM</sup>) to retrieve articles focused on HTPP using sUAS platforms. The keywords used during this search included "UAV", "UAS", "Drone", "unmanned aerial vehicle", "unoccupied aerial vehicle", "unmanned aerial system", "unoccupied aerial system", "phenotyping", "phenomics", "multispectral", "hyperspectral", "sensors", "LiDAR", "crop", and "plant". An additional search was conducted using Google Scholar with the same keywords. Although the application of sUAS in HTPP is relatively new, all literature searches were conducted considering a timeframe within the last three decades (1990–2021). Over this period, the literature search revealed 205 research

articles with an emphasis on HTPP or field-based crop trait estimation. In 34% of the total retrieved articles, traits associated with plant growth (such as plant height, leaf area index, and canopy cover) were evaluated. Other traits of interest included salinity resilience, disease resistance, nutrient and water use, and crop yield, among others.

Early works on the application of the sUAS-based imagery in agriculture were focused on crop management rather than HTPP [64,65], and these works were not included in this review. Additionally, there were no sUAS-based HTPP studies reported prior to 2009 (Figure 1). The number of published sUAS-based articles with a focus on HTPP significantly increased from 7 in 2015 to 92 in 2021, suggesting an increased application of sUAS-based imagery in HTPP. A list of journals with at least three published articles on field-based sUAS crop phenotyping is presented in Table 1. Totals of 29%, 12%, and 10% of articles retrieved were published in *Remote Sensing*, *Frontiers in Plant Science*, and *Computers and Electronics in Agriculture*, respectively.

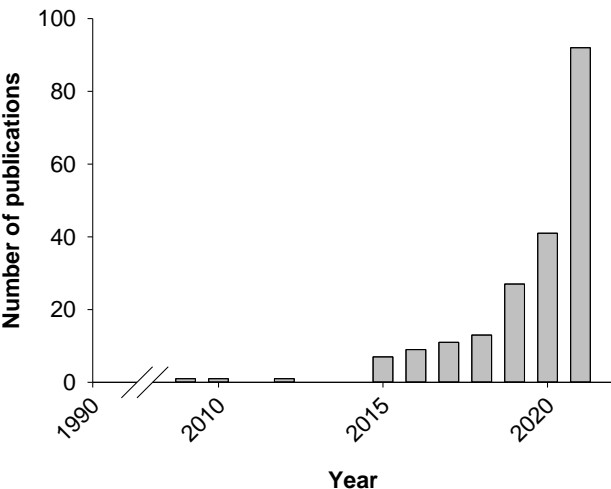

**Figure 1.** Number of peer-reviewed articles related to sUAS-based HTPP (1990–October 2021).

**Table 1.** Relevant journals with at least three published articles related to sUAS-based HTPP (1990–October 2021).

| Journals | Number of Publications | Percentage of Total Publication |
|---|---|---|
| *Remote Sensing* | 59 | 29 |
| *Frontiers in Plant Science* | 25 | 12 |
| *Computers and Electronics in Agriculture* | 20 | 10 |
| *Field Crops Research* | 12 | 6 |
| *Plant Methods* | 11 | 5 |
| *Sensors* | 9 | 4 |
| *Journal of Experimental Botany* | 8 | 4 |
| *Agronomy-Basel* | 5 | 2 |
| *ISPRS Journal of Photogrammetry and Remote Sensing* | 4 | 2 |
| *Remote Sensing of Environment* | 4 | 2 |
| *IEEE Access* | 4 | 2 |
| *PLOS One* | 3 | 1 |
| *Scientific Reports* | 3 | 1 |
| *Precision Agriculture* | 3 | 1 |
| Others | 35 | 17 |
| Total | 205 | 100 |

## 2. Common sUAS-Based Imaging Techniques in HTPP

Electronic sensors are the primary components of data acquisition in sUAS-based HTPP. Such sensors can be classified into two categories: auxiliary and phenotyping.

Auxiliary sensors ensure that flights are completed safely and efficiently and keep sUAS in a balanced position during flight [66]. Phenotyping sensors are imaging sensors aboard the sUAS used primarily for measuring or estimating crop phenotypes [25]. The common types of sUAS imaging sensors are classified as follows: red-green-blue (RGB), multispectral, hyperspectral, and thermal infrared [25]. Light detection and ranging (LiDAR) systems are also used onboard sUAS and can be used for building detailed 3-dimensional surface models [67].

### 2.1. Color Imaging

Images from an RGB color camera are intended to mimic human perception. Digital color cameras are relatively inexpensive and ubiquitous, which has improved their practicality for use in HTPP [68]. Although a standard digital color camera uses silicon-based sensors that are sensitive to light from 400 to 1000 nm [68], most cameras filter the light to 400–750 nm [20], thereby removing the near-infrared signal from the measurements. The cameras also typically incorporate a color filtering system that measures photon fluxes of red, green, and blue spectral bands at approximately 600 nm, 550 nm, and 450 nm, respectively. Human perception of color in these images is dictated via the levels of visible red, green, and blue light recorded by the camera, and the quantification of photon flux in the three channels serves as the basis for a variety of computational approaches for plant trait estimation.

Owing to its relatively low cost and ease of operation, the color image has been widely used for HTPP both in controlled environments and in the field. Current applications of color imaging in HTPP include the estimation of plant or canopy height, canopy cover, leaf area index (LAI), crop water use, lodging, biotic and abiotic stress response, biomass production, and seedling emergence [4,53,69–71]. Although visible imaging has been applied to directly measure or estimate many phenotypes for HTPP, the technology may be impractical for other traits, such as leaf area, leaf number, or biomass, due to the occlusion of the lower canopy components by the upper canopy [20]. In other words, visible imaging technology does not penetrate the canopy, and imaging focuses mainly on the characteristics of the plant, canopy, and soil surfaces. The application of visible imaging in HTPP can also be limited by the shadowing of some canopy features by others and field conditions with low contrast between the target of interest and the surrounding background [20].

### 2.2. Thermal Imaging

Thermal imaging involves the visualization of infrared radiation as an indication of temperature distributions across the body of an object [20,72]. Thermal cameras are sensitive within a spectral range of 3–14 μm and are commonly used within the wavelengths of 3–5 μm or 7–14 μm. The transmission of solar infrared radiation in the Earth's atmosphere is close to its maximum value within these two ranges. Smaller wavelengths correspond to greater energy levels; hence, their thermal sensitivity at the shorter wavelengths (3–5 μm) is greater than that of the longer wavelengths (7–14 μm).

Thermal imaging can be applied to measure leaf or canopy temperatures, which is especially important in the study of plant water relations as affected by stomatal conductance [20]. This is because the rate of stomatal conductance is a major factor that determines leaf temperature, especially under reduced soil moisture conditions [20,73]. Plants often respond to reduced soil moisture conditions by closing leaf stomata to conserve water, thereby reducing the rate of photosynthesis ($CO_2$ in) and transpiration ($H_2O$ out) [74,75]. A reduced rate of transpiration has been shown to increase leaf temperature (due to reduced effects of evaporative cooling) under water stress conditions [76,77]. Therefore, thermal imaging can be used to estimate plant water stress [78,79]. Thermal cameras are more costly and more difficult to deploy compared to infrared thermometers; however, thermal cameras have higher spatial resolution under changing environmental conditions [20].

### 2.3. Imaging Spectroscopy

Plant imaging spectroscopy measures interactions of radiation with vegetation. Due to the effects of plant chemical composition and canopy architecture on the spectral reflectance properties of vegetation, spectral response patterns across wavelengths can be used to estimate crop traits expressed under different genetic, environmental, or management conditions [80,81]. Spectral reflectance of single leaves as well as plant canopies is reduced within the visible spectrum (400–700 nm) due to light absorption by leaf pigments (principally chlorophyll). The spectral response of vegetation typically demonstrates a reflectance peak in the visible green region of the spectrum around 550 nm, which causes the appearance of green color to the human eye. At the transition between the visible and near-infrared (NIR) wavelengths, a sharp increase in leaf reflectance, known as the "red edge," occurs between 690 and 740 nm, and the reflectance of NIR from vegetation remains largely up to 1200 nm [20]. Because NIR radiation is easily transmitted within the layers of the plant canopy, leaf thickness, and canopy architecture are key factors that drive vegetative spectral properties in this part of the spectrum. From 1200 to 2500 nm, the spectral reflectance of vegetation is governed by the water content present within the leaf tissue, and several distinct absorption bands appear at 1450 and 1920 nm [82,83]. However, when relaying on solar illumination, absorption of radiation by atmospheric water vapor at 1380 nm and 1870 nm can drastically reduce signal at these wavelengths, which can impede the analysis of the water absorption feature of vegetation spectra. The spectral reflectance of leaves or plant canopies at specific wavelengths can be used to compute spectral vegetation indices, such as the normalized difference vegetation index (NDVI). The NDVI and a host of other indices have been routinely tested for use in HTPP [20]. Furthermore, due to the wealth of information present in spectral reflectance data, it serves as a useful data source for modern studies in data mining and machine learning, which can identify the most useful metrics for associating spectral information with plant traits of interest [20,84].

Imaging spectroscopy from sUAS has been applied in HTPP for the estimation of plant biomass, photosynthetic efficiency, water content, greenness, nitrogen composition, and disease presence [50,52,70,85–87]. For example, terrestrial-based multispectral and hyperspectral measurements have been used to estimate crop water content [88–90], while Chivasa et al. [49] used sUAS-based multispectral imaging for phenotyping streak virus disease in maize. While RGB cameras measure spectral information in only three broad wavebands in the visible spectrum, hyperspectral sensors can measure reflectance in hundreds of narrow, continuous spectral bands [91]. Hence, hyperspectral imaging provides a greater opportunity to identify important spectral metrics for better trait identification in HTPP. However, image acquisition via hyperspectral sensors may take longer compared to digital color sensors [28,92–95]. Additionally, the specialized and complex nature of hyperspectral sensors leads to greater costs (~$40,000 USD), and the weight is often above 1 kg [96]. Thus, given the sensor cost and the limited payload capacity of sUAS, the application of hyperspectral sensors in HTPP may require advanced sUAS capability. Finally, due to the high dimensionality of hyperspectral data, greater computational capabilities and data management resources are required for the use of hyperspectral data in HTPP compared to color images [97,98].

### 2.4. Light Detection and Ranging (LiDAR) Imaging

Light detection and ranging (LiDAR) refers to a laser-based sensor that produces high-density, three-dimensional point clouds via photon counting [99]. The fundamentals of LiDAR sensors are based on the time-of-flight (TOF) principle. The sensors send light pulses toward an object and receive reflected pulses from the object [53]. The distance between the sensor and the object is, therefore, calculated as half of the product of the speed of light and the time required for light pulses to be sent and received [100]. Over the years, the TOF principle has been increasingly applied in several aspects of precision agriculture due to its accuracy, versatility, and reading speed [100]. LiDAR operational capacity is

robust and can be used under a wide range of light conditions [101,102], including at night [45]. Therefore, LiDAR sensors are a promising technology for measuring plant height and other architectural traits [103].

While sUAS-based LiDAR imaging has previously been used in forestry [104,105], few studies on the application of sUAS-based LiDAR imaging have been conducted for HTPP in agronomic crops. Harkel et al. [106] evaluated sUAS-based LiDAR for the estimation of plant height and biomass in winter wheat, sugar beet, and potato crops. In this study, the LiDAR sensor worked well to estimate both plant height and biomass in winter wheat, but it was less accurate for plant height and biomass estimation in potato and sugar beet. Compared to winter wheat with an erectophile structure and homogenous pattern in height, potato and sugar beet have planophile structures with complex leaf angle distributions. Therefore, accurate plant height estimation under complex canopy structures may be more challenging [106]. Other studies have also used LiDAR sensors for estimating plant biomass and height in maize [107,108] and cotton [53].

## 3. sUAS-Based Image Processing and Data Analysis

Prior to feature or phenotype extraction in HTPP, sUAS-based images are subjected to several processing steps to improve and enhance the image data representation. Part of the critical initial processing steps after image capturing are image alignment and stitching to generate a georeferenced orthomosaic image [109]. Leading proprietary software programs often used for sUAS image alignment and stitching include Agisoft Photoscan (Agisoft), Pix4Dmapper (Pix4D), and OpenDroneMap (ODM). These programs are well known among HTPP researchers as they have been used in several research studies [49,53,69,110,111]. Unlike Pix4D and Agisoft, ODM is a free and open-source photogrammetry software.

The ultimate goal of these programs as applied to HTPP is to use multiple still-frame sUAS images to build 3D digital surface models (DSM) and produce 2D orthomosaics of the field area. Image processing using Agisoft involves four main stages, including alignment, generation of the dense point cloud, generation of the surface model (i.e., mesh and/or digital elevation model (DEM)), and surface reconstruction. Photo alignment using Agisoft is based on camera orientation at the time of image capture [112]. The camera orientation is a function of the interior (focal length, coordinates, and lens distortion coefficients) and exterior (position and orientation of the camera) parameters of the camera at the time of image capture. Based on the estimated interior and exterior parameters, the program calculates the depth maps that are transformed into partial dense point clouds, which are then merged into the final dense point cloud. The resulting dense point cloud is used to develop a DSM, digital terrain model (DTM), and orthomosaic [112]. Image analysis using OpenDroneMap (ODM) and Pix4D generally follows a similar approach as described for Agisoft [113]. Both Agisoft and ODM use a Structure from Motion (SfM) library (OpenSfM, Mapillary AB, Malmo, Sweden) for 3D point cloud construction; thus, camera calibration (position and orientation) is automatically solved or refined during processing [114]. Unlike Agisoft and ODM, Pix4D does not use SfM but uses a machine vision algorithm through an automated feature detection and matching process. Although these photogrammetry programs may differ slightly in their backend (especially Pix4D) process, their workflow and end products are generally similar. There are many other photogrammetry programs, including DroneDeploy, PhotoCatch, OpenMVG, and VisualSFM; however, their applications in HTPP are either limited or nonexistent.

Depending on the feature or phenotype of interest, the resulting products from these programs are often subjected to further analyses prior to feature extraction, including classification, regression, or clustering [45]. Crop traits such as plant height, LAI, and canopy cover can be directly extracted from processed images, while traits such as plant nutrient and water content, chlorophyll content, and yield are often acquired in relation to vegetation indices using empirical statistical models. The commonly adopted statistical methods include multiple linear, stepwise linear, and partial least square linear regression

models [115]. With the increase in technology and knowledge in data science, advanced data analysis and machine learning such as convolutional neural networks (CNNs) are important methods for improving prediction accuracy in HTPP. CNNs are the most widely adopted deep-learning approach for image recognition [87]; however, they require large amounts of data to produce meaningful output. The operation of these networks is time-consuming and requires complex data processing and analysis; hence, application in HTPP may be more challenging [116,117].

## 4. Common sUAS-Based Trait Extraction for HTPP

Imagery from sUAS has been extensively used to evaluate a wide range of traits for growth (plant height, canopy cover, leaf area index, etc.), biotic and abiotic stress resilience (diseases, heat stress, drought stress, salinity, etc.), and resource-use efficiency (nutrient recovery, water use, radiation use, etc.), among others (Table 2). This section summarizes several HTPP-based traits or phenotypes extracted using sUAS platforms.

### 4.1. Plant Growth

Traits such as canopy cover and height, leaf area index (LAI), and biomass have been evaluated in grain, fruit, and energy crops. Compared to other trait categories, crop growth traits can be rapidly estimated using all types of sensors aboard sUAS. Plant growth traits (Table 2) have been estimated more often (especially for breeding purposes) using sUAS-based imagery compared to other traits reported in the literature because these traits are correlated with crop development and yield [118]. For example, plant height has often been reported as positively correlated with fiber yield and water-use efficiency under low soil moisture conditions in cotton [119,120]. Although sUAS-based plant height phenotyping was reported for blueberry in 2017 [121], this trait has mostly been estimated in row crops such as cotton, soybean, maize, and sorghum [52,54,69,85,122–124]. Early works on open-field sUAS-based growth trait estimation were reported in 2015 by Sankaran et al. [125] during the winter season in Kahlotus Washington, USA. Although the study focused on evaluating seed emergence and growth in winter wheat using multispectral sensors, a later study by Holman et al. [126] in 2016 evaluated the plant height and growth rate of winter wheat in Rothamsted, UK. In this study, plant height and growth rate were estimated from a multi-temporal, high spatial resolution (1 cm pixel$^{-1}$), 3D DSM created from sUAS-based RGB images. More recent studies have applied crop image analysis procedures to estimate plant height for multiple crops (including fiber, grain, and vegetable crops) using different imaging sensors (RGB, multispectral, and LiDAR) [53,127–131]. For example, Maesano et al. [102] reported a significant positive relationship between manual and LiDAR-based measurement of plant height and biomass production in bioenergy grass (*Arundo donax*).

In addition to plant height measurements, LAI is another growth trait commonly evaluated using sUAS platforms in crops such as maize, soybean, blueberry, and rice [50,70,121,122]. LAI is an indicator of the size of the assimilatory surface of a crop [132]; hence, it is often described as one of the main driving forces for net primary production, carbon balances, and water- and nutrient-use efficiencies [133]. Fenghua et al. [51] used a radiative transfer model with hyperspectral images to estimate rice canopy properties (including LAI) from crop spectra data. Similarly, Su et al. [122] used a radiative transfer model to estimate LAI in maize. Maimaitijiang et al. [50] used the fusion of multiple sensors (RGB, multispectral, and thermal) to estimate both biochemical and biophysical traits (including LAI) in soybean. Other growth traits, such as canopy cover and plant biomass accumulation, have also been evaluated using sUAS-based hyperspectral and multispectral imagery [50,52,57,69,70,134]. Based on the information gathered from the literature, it is evident that nearly all sensors (RGB, multispectral, hyperspectral, and LiDAR) can be effectively used to estimate plant growth traits with high correlation (r$^2$ ranging from 0.57 to 0.98) between manual and sensor-based measurements.

**Table 2.** Selected crop traits evaluated from the analysis of crop images captured using sUAS-based platforms for HTPP, as found in the literature.

| Crop | Sensor | | | | | Traits | | | | | | | | | | | | Source |
|---|---|---|---|---|---|---|---|---|---|---|---|---|---|---|---|---|---|---|
| | RGB | Multi-Spectral | Hyper-Spectral | Thermal | LiDAR | Plant Height | Canopy Cover | Leaf Area Index | Biomass | Salinity Stress | Drought Stress | Nutrient Stress | NUE, WUE | Diseases | Yield | Veg. Index | Other Traits | |
| Soybean | X | | | | | X | X | | | | | | | | | | | Borra-Serrano et al. [69] |
| | X | X | | X | | | | | | X | X | | | | | | | Maimaitijiang et al. [50] |
| | | | | X | | | | | | | | X | | | | | | Sagan et al. [55] |
| Corn | X | X | | | | X | | X | | | | | | | | | X | Su et al. [122] |
| | X | | X | | | X | | | X | | | | | | | | X | Yang et al. [123] |
| | X | X | | | | X | X | | | | | | | | | | X | Han et al. [85] |
| | X | | | | | X | | | | | | | | | | | | Wang et al. [135] |
| | | X | | | | | | | | | | | | | X | | | Chivasa et al. [49] |
| | X | | | | | | | | | | | | | | X | | | Stewart et al. [136] |
| Cotton | | X | | | | X | X | | | | | | | | | X | | Xu et al. [52] |
| | X | | | | | X | | | | | | | | | | | | Thompson et al. [53] |
| | | X | | | | X | X | | | | | | | | | X | X | Xu et al. [52] |
| | | X | | | | | | | | | | | X | | | | | Thorp et al. [4] |
| Wheat | | X | | | | | | | | | | | X | | | | | Yang et al. [86] |
| | | | | X | | | | | | | | | X | | | | | Perich et al. [111] |
| | | | X | | | | | | | | | | | | | | | Moghimi et al. [56] |
| | X | X | | X | | | | | | | | | X | | | | | Gracia-Romero et al. [137] |
| | | | X | | | | | | | | | | | | | | X | Sankaran et al. [125] |
| | | | X | | | | | | | | | | | | | | X | Camino et al. [138] |
| Wheat | | X | | X | | | | | | | | | | | | X | | Gonzalez-Dugo et al. [139] |
| | | X | | | | | | | X | | | | | | X | | | Ostos-Garrido et al. [57] |

**Table 2.** *Cont.*

| Crop | Sensor | | | | | Traits | | | | | | | | | | | | Source |
|---|---|---|---|---|---|---|---|---|---|---|---|---|---|---|---|---|---|---|
| | RGB | Multi-Spectral | Hyper-Spectral | Thermal | LiDAR | Plant Height | Canopy Cover | Leaf Area Index | Biomass | Salinity Stress | Drought Stress | Nutrient Stress | NUE, WUE | Diseases | Yield | Veg. Index | Other Traits | |
| Sorghum | X | | | | | X | | | | | | | | | | | | Hu et al. [124] |
| | | | | X | | | | | | | X | | | | | | | Sagan et al. [55] |
| Barley | | X | | | | | | | | X | | | | | | X | | Ostos-Garrido et al. [57] |
| | X | X | | X | | | | | | | | | X | | | | | Kefauver et al. [110] |
| Dry bean | | X | | X | | | | | X | | | | | | | | | Sankaran et al. [51] |
| | | X | | | | | | | | | | | | X | | X | | Sankaran et al. [134] |
| Rice | | | X | | | | | X | X | | | | X | | | | X | Fenghua et al. [70] |
| Potato | X | | | | | | | | | | | | | X | | | | Sugiura et al. [140] |
| Blueberry | X | | | | | X | X | | | | | | | | | | | Patrick and Li [121] |
| Peanut | X | | | | | | | | | | | | | X | | | | Patrick et al. [141] |
| Citrus | | X | | | | | X | | | | | | | X | | | | Ampatzidis and Patel [87] |
| Tomato | X | | | | X | | | | | | | X | | | | | | Johansen et al. [142] |
| Sugar beet | | | | | X | X | | | X | | | | | | | | | Harkel et al. [106] |
| Bioenergy crop | | | | | X | X | | | X | | | | | | | | | Maesano et al. [102] |

## 4.2. Abiotic Stress Resilience and Adaptation

The application of sUAS technology can greatly contribute to the ongoing global investigations on identifying resistant cultivars and traits associated with adaptation to various stress conditions. For instance, RGB, thermal, and multispectral sUAS-based imagery have recently been applied for HTPP in tomato [142], dry bean [51,134], soybean, and sorghum [55] to determine plant responses to salinity and drought stress conditions. Sankaran et al. [134] reported that both canopy area and green normalized difference vegetation index (GNDVI) were significantly ($p < 0.05$) correlated with seed yield and biomass production of dry beans under drought conditions. Hence, GNDVI could be a viable indicator of both the yield and biomass of dry beans under drought conditions. This study also found that differences in stress-induced canopy temperature could also be measured using thermal imaging. A more recent study on dry beans indicated that the normalized difference vegetation index (NDVI) was found to be strongly and consistently correlated with above-ground biomass and yield [51] under multiple stress conditions. Similarly, Johansen et al. [142] were able to use sUAS-based RGB and multispectral imagery to identify high-yielding accessions in tomatoes under salt stress [142], while thermal imaging was used to determine the relationship between water stress and canopy temperature in soybean and sorghum [55]. Unlike plant growth traits that are often extracted directly from processed images, estimation of traits associated with crop responses to abiotic stress resilience and adaptation in HTPP are heavily reliant on indirect traits such as vegetation indices. Thus, spectral imaging sensors have an important role in the development of prediction models for abiotic stress detection in crops.

## 4.3. Nutrient- and Water-Use Efficiencies and Crop Yield

Among the goals of any breeding program are efficient resource use and yield increases. Increases in water- and nutrient-use efficiency are critical for reducing nutrient loss to the environment and for improving water productivity. The development of improved cultivars with high-yielding genetic potential has been identified as a possible pathway for improving crop water-use efficiency [143]. Therefore, HTPP with a focus on identifying indirect traits for improved yield and nutrient and water use efficiencies is considered essential. Compared to the time-consuming, costly, and destructive efforts associated with manual sampling, sensor technologies aboard sUAS have been efficiently used to identify traits associated with improved yield and water and nutrient use efficiencies in crop production. For example, Yang et al. [86] reported that the normalized red-green difference index (NGRDI) was the most efficient non-destructive indicator for water use-efficiency calculated from biomass production ($r^2$ between 0.69–0.89) and grain yield ($r^2$ between 0.80–0.86) in winter wheat. These authors also demonstrated that red normalized difference vegetation index (RNDVI) could be used for estimating nitrogen (N) use efficiency ($r^2$ up to 0.94), calculated from the tissue N content in wheat, while normalized difference red-edge index (NDRE) was most consistent ($r^2$ up to 0.84) in predicting nitrogen use-efficiency calculated from observed yield. Similarly, Thorp et al. [4] used weekly vegetation cover estimates from a drone-based multispectral camera to develop basal crop coefficients for a daily water balance model. The model was able to determine differences in water use estimates among several cotton cultivars. Therefore, combining water use estimates from this model with observed yield could be a more effective way to improve the selection of cultivars with favorable water use characteristics. Similarly, Gracia-Romero et al. [137] showed that the proportion and duration of green biomass involved in the partitioning of photosynthate to grain, as well as crop vigor, were important traits for selecting wheat germplasm with improved adaptability (in terms of improved crop growth and yield) for the Mediterranean environment. Other studies by Hu et al. [144] and Moghimi et al. [56] also used sUAS-based imagery to determine traits associated with grain yield in wheat and obtained prediction accuracy (quantified using $r^2$) from 0.60 to 0.79. With the rapid advancement in the application of sUAS-based remote sensing for HTPP, the literature search suggested that sUAS offer a promising, rapid, reliable, and cost-effective approach

to enhance crop breeding for greater productivity and water- and nutrient-use efficiencies in crop production. Similar to abiotic stress and yield predictions, the literature search also suggested that spectral sensors play essential roles in the estimation of crop water and nutrient contents. Other sensors, such as LiDAR and thermal, may not be effective for estimating plant nutrient content.

*4.4. Disease Detection and Crop Resilience to Biotic Stress*

The genetic potential of any crop variety may not be fully realized if adequate provision is not made for disease management. As a result, routine monitoring of crop diseases is a critical production practice carried out by growers to reduce the risk of both production and economic losses [145]. sUAS-based imagery with fine spectral resolution (especially hyperspectral imaging at less than 10 nm intervals) can be used to detect early disease symptoms for timely decision making on the implementation of effective control measures [93,146]. Previous studies used sUAS-based platforms for monitoring, detecting, and evaluating crop responses to disease severity, as well as the identification of disease-resistant traits in agricultural crops (Table 2).

Chivasa et al. [49] used sUAS-derived spectral data obtained from a multispectral sensor to determine the severity of maize streak virus disease in 25 maize varieties. The authors reported a good overall classification accuracy (77%) for resistant, moderately resistant, and susceptible cultivars using a Random Forest classifier. The authors also indicated that vegetation indices such as green normalized difference vegetation index (GNDVI), green chlorophyll index ($CI_{green}$), red-edge chlorophyll index ($CI_{rededge}$), and the red band were more important for maize streak virus classification compared to the normalized difference vegetation index (NDVI) and green normalized difference vegetation index (GNDVI). Stewart et al. [136] used a masked region-based convolutional neural network (R-CNN) to detect and segment northern leaf blight (NLB) disease lesions in maize. In this study, the precision of the classification procedure was estimated based on an established intersection-over-union (IOU) threshold. The mean IOU is a common evaluation metric for semantic image segmentation, which first estimates the IOU for each class and then computes the average over all classes [147]. The authors reported that R-CNN was able to detect NLB disease with up to 96% accuracy. Although due to a higher image resolution requirement for in-field sUAS-based plant disease detection and classification, most studies are often conducted using hyperspectral and multispectral sensors [49,136,141]; however, low-cost RGB sensors have been used with acceptable results ($r^2 = 0.77$) in estimating late blight disease severity in potato [140]. Overall, RGB, multispectral, and hyperspectral imagery have been successfully used in the monitoring, identification, and classification of crop diseases. However, RGB sensors are often characterized by lower disease detection accuracy compared to multispectral and hyperspectral sensors. Generally, sUAS technology presents a valuable opportunity for quick identification and classification of crop diseases for breeding selection as well as early disease detection for a timely intervention.

*4.5. Other Areas of Application*

Several other areas of applications of sUAS-based platforms for HTPP were also found during the literature search. For example, Lopez-Granados [148] used an RGB camera onboard an sUAS to monitor flowering dynamics and create a flowering calendar for almond tree crops. Ostos-Garridos [57] used sUAS-based multispectral imagery to determine bioethanol production in wheat, barley, and triticale, and Han et al. [123] used RGB and multispectral imagery to determine lodging in maize. Ampatzidis and Partel [87] determined tree health and tree count using a deep-learning CNN in citrus. The study reported 99.9% and 85.5% overall accuracy for tree count and canopy area detection, respectively, compared to manual measurements. Other applications included the estimation of plant chlorophyll, water content, and seedling emergence [50,70,125,138]. Maimaitijiang et al. [50] reported that tissue chlorophyll-a content and nitrogen concentration were best described (RMSE of 9.9% and 17.1%, respectively) via multispectral and thermal data fusion.

However, chlorophyll a + b was best described using data fusion from RGB, multispectral, and thermal sensors. In summary, sUAS-based imagery has been successfully and efficiently applied for HTPP over a wide range of application areas in several crops and growing conditions.

## 5. Comparing Crop Trait Estimation from Imaging Sensors on Terrestrial versus sUAS Platforms

The accuracy of trait extraction in HTPP research is often dependent on the crop and sensor type, the complexity of the trait of interest, and the quality of the image processing. Therefore, the limitations as well as the suitability of various imaging sensors and platforms (terrestrial versus sUAS) are described below, based on their accuracies in trait estimation, as reported in the literature. This analysis is focused on synthesizing results from studies on crop growth traits (i.e., height, biomass production, canopy cover, and LAI) and yield (Table 3).

**Table 3.** Comparison of trait estimation accuracies based on platform, crop, and imaging sensors.

| Crop Phenotype or Trait | HTPP Platform | Crop | Sensor | Evaluation Method | | | Reference |
|---|---|---|---|---|---|---|---|
| | | | | $r^2$ | RMSE | Accuracy | |
| Plant height | Aerial (sUAS) | Cotton | RGB | 0.98 | | | [53] |
| | | | Multispectral | 0.90–0.96 | 5.50–10.10% | | [52] |
| | | | | 0.78 | | | [122] |
| | | Soybean | RGB | 0.70 | | | [69] |
| | | | | >0.70 | | | [69] |
| | | Maize | RGB | 0.78 | 0.168 | | [122] |
| | | | | 0.95 | | | [135] |
| | | Rice | | 0.71 | | | [54] |
| | | Sorghum | RGB | 0.57–0.62 | | | [149] |
| | | | NIR-GB | 0.58–0.62 | | | |
| | | | RGB | 0.69–0.73 | | | |
| | | | | 0.34 | | | [124] |
| | | Bioenergy grass (*Arundo donax*) | LiDAR | 0.73 | | | [102] |
| | Terrestrial or ground platform | Peanut | RGB | 0.95 | | | [150] |
| | | | | 0.86 | | | [150] |
| | | Blueberry | RGB | 0.92 | | | [121] |
| | | Maize | Laser scanner | 0.93 | | | [151] |
| | | | Ultrasonic | 0.87 | 3.10 cm | | [152] |
| | | Cotton | RGB-D | 0.99 | 0.34 cm | | [153] |
| | | | | 0.97 | | | [154] |
| | | | LiDAR | 0.98 | 6.50 cm | | [155] |
| | | Triticale | RGB | 0.97 | | | [156] |
| | | | | 0.86 | 7.90 cm | | [24] |
| | | Wheat | LiDAR | 0.90 | | | [157] |
| | | | | 0.99 | 1.70 cm | | [158] |
| | | | RGB | 0.95 | 3.95 cm | | [159] |
| | | Sorghum | Ultrasonic | 0.93 | | | [160] |
| | | | LiDAR | 0.88–0.9 | | | |

**Table 3.** *Cont.*

| Crop Phenotype or Trait | HTPP Platform | Crop | Sensor | Evaluation Method | | | Reference |
|---|---|---|---|---|---|---|---|
| | | | | r² | RMSE | Accuracy | |
| Canopy cover and leaf area index | sUAS | Blueberry | RGB | 0.70–0.83 | | | [121] |
| | | Soybean | RGB | >0.70 | | | [69] |
| | | Cotton | Multispectral | 0.33–0.57 | | | [52] |
| | | Citrus | Multispectral | | | 0.85% | [87] |
| | | Rice | Hyperspectral | 0.82 | 0.10 | | [70] |
| | | Soybean | RGB and Multispectral fusion | | 0.059 | | [50] |
| | Terrestrial or ground platform | Maize | RGB | 0.75 | 0.34 | | [122] |
| | | Cotton | LiDAR | 0.97 | | | [154] |
| | | Wheat | LiDAR | 0.92 | | | [158] |
| | | Soybean | RGB | 0.89 | | | [161] |
| | | | HSI | 0.80 | | | [161] |
| | | Maize | LiDAR | 0.92 | | | [162] |
| | | Sorghum | LiDAR | 0.94 | | | [162] |
| Biomass or dry matter production | sUAS | Soybean | Multispectral and thermal fusion | | 0.10 | | [50] |
| | | Rice | Hyperspectral | 0.79 | 0.11 | | [70] |
| | | Barley, triticale, and wheat | Multispectral | 0.44–0.59 | | | [57] |
| | | Maize | Hyperspectral | 0.47 | | | [108] |
| | | | LiDAR | 0.83 | | | |
| | | | Hyperspectral and LiDAR fusion | 0.88 | | | |
| | | Dry bean | Multispectral and thermal | (−0.67)–(−0.91) | | | [51] |
| | Terrestrial or ground platform | Bioenergy grass (*Arundo donax*) | LiDAR | 0.71 | | | [102] |
| | | Wheat | LiDAR | 0.92–0.93 | | | [158] |
| | | Sugar beet | RGB | 0.82–0.88 | | | [163] |
| | | Cassava | LiDAR | 0.73 | | | [164] |
| | | Maize | LiDAR | 0.68–0.80 | | | [165] |
| Canopy temperature and yield | sUAS | Wheat | RGB | 0.94 | | | [166] |
| | | | | 0.60 | | | [144] |
| | | | Multispectral | 0.63 | | | [144] |
| | | | | 0.65 | | | |
| | | | | 0.43 | | | |
| | | | | 0.57 | | | |
| | | Rice | RGB | 0.73–076 | | | [167] |
| | | | Multispectral | 0.82 | | | [144] |
| | Terrestrial or ground platform | Wheat | Multispectral and thermal | 0.52 | | | [168] |
| | | | | 0.36 | | | |
| | | | | 0.51 | | | |

### 5.1. Plant Height Estimation

Plant height is an important trait for evaluating plant growth and can be used as an indicator of biomass production, lodging, yield, and water and nutrient uptake [53]. Plant height is well studied under both terrestrial and sUAS platforms, and the accuracy of data extraction from crop images varies among studies. RGB sensors were commonly used for plant height estimation in sUAS-based HTPP research, possibly due to their low cost [25]

and reduced weight [169]. On the other hand, LiDAR sensors were more commonly used for plant height estimation on ground-based platforms (Table 3).

For sUAS-based plant height detection, Thompson et al. [53] used 3D point cloud data obtained from open-source photogrammetry software (OpenDroneMap) to determine plant height from sUAS-based RGB images in cotton. Operations of the software were based on an SfM library to perform feature extraction matching and 3D point cloud computation. The authors reported a high prediction accuracy ($r^2$ = 0.98) compared to the ground truth, and the accuracy was similar to those observed from a terrestrial ultrasonic sensor. A lower correlation ($r^2$ = 0.78) was reported by Su et al. [122] using a DSM model obtained from SfM photogrammetry methods for the estimation of plant height in cotton. Hansen et al. [170] obtained a high correlation ($r^2$ ranging from 0.80 to 0.85) between sUAS-based RGB imagery of plant height and manual measurements in wheat during booting and grain filling stages. Similarly, RGB-based ground point data yielded a higher level of prediction accuracy ($r^2$ = 0.95) for plant height compared to an $r^2$ value of 0.86 using a digital terrain model (DTM) in peanuts [150]. However, the prediction accuracy reported by Borra-Serrano et al. [69] for plant height obtained from sUAS-based RGB images was lower ($r^2$ = 0.70) with a DEM-based model in soybean. A slightly lower prediction accuracy ($r^2$ = 0.63) for plant height was also reported by Hu et al. [124] using a proposed self-calibration model on sUAS-based RGB images in sorghum. However, these authors reported that the proposed model performed better than the DSM ($r^2$ = 0.34) model. The self-calibration method was proposed as an alternative plant height estimator when the soil surface between plots was not visible (i.e., during maximum canopy cover). A similar range of accuracy ($r^2$ = 0.57–0.73) was reported for plant height extraction from either RGB or NIR-GB images in other studies on sorghum [124,149] and rice [54]. Other than RGB sensors, Xu et al. [52] reported that maximum cotton height estimated using a DEM-based model from a multispectral sensor aboard an sUAS was highly correlated ($r^2$ = 0.90–0.96, RMSE = 5.5%–10.1%) with manual height measurements. However, a lower level of accuracy ($r^2$ = 0.71–0.75) was reported by Maesano et al. [102] using an sUAS-based LiDAR sensor for estimating plant height in a bioenergy crop (*Arundo donax*). A similar range of accuracy was reported using an airborne LiDAR sensor in sugar beet ($r^2$ = 0.70) and wheat ($r^2$ = 0.78) [106].

Similar to the sUAS platform, HTPP studies with a focus on plant height measurement using terrestrial platforms were also evaluated over multiple crops using different sensing technologies. Tilly et al. [151] used point cloud data obtained from a terrestrial laser scanner to estimate plant height in maize with a high correlation ($r^2$ = 0.93) reported between the estimated sUAS-based plant height and manual measurements. Several other terrestrial-based HTPP studies similarly obtained high plant-height estimation accuracy in several other crops, including cotton ($r^2$ = 0.87) using an ultrasonic sensor [152], triticale ($r^2$ = 0.97) using an RGB sensor [156], wheat ($r^2$ = 0.86–0.99) using LiDAR [157] and RGB ($r^2$ = 0.95) [159] sensors, and sorghum ($r^2$ = 0.93) using ultrasonic and LiDAR ($r^2$ = 0.88–0.99) sensors [160].

Based on the results from the literature retrieved, the reported accuracies of plant height estimation were generally similar for both the sUAS and terrestrial platforms; however, the use of LiDAR sensors aboard sUAS platforms for plant height estimation was somewhat less accurate compared to RGB sensors. Although LiDAR sensors aboard sUAS platforms are common for estimating height metrics and laser penetration indices under forest conditions [171–173], relatively lower prediction accuracy was found for sUAS-based LiDAR height estimation in crops. This could possibly be because (1) short dense vegetation limits laser penetration to the ground [174,175], and (2) measurements are collected using discrete LiDAR compared to a full waveform LiDAR system. Discrete LiDAR sensors only record limited returns for each emitted pulse hence data from discrete-LiDAR sensors may provide limited information about vertical vegetation structures compared to a full waveform LiDAR [174–176]. Few studies directly compared both platforms (sUAS and terrestrial) for plant height, and the methodologies generally favored the use of RGB sensors aboard sUAS and LiDAR on terrestrial platforms [53,157]. Other than RGB and LiDAR,

plant height data from both multispectral and ultrasonic sensors were reported to be highly correlated with manual measurements for both sUAS and terrestrial platforms [52,53,152]. Therefore, considering sensor cost, weight (especially for sUAS platforms), data processing complexities, and sensor accuracy for plant height estimation, RGB sensors are adequate for estimating plant height from both sUAS and terrestrial HTPP platforms.

Similar to sensor type and platform, the information from the literature also indicated that the accuracy of plant height estimation in HTPP can be influenced by the image processing model of choice. Generally, RGB-based plant height measurements using only 3D point cloud data derived from the sUAS platform are more accurate ($r^2$ 0.95–0.98) compared to height estimation from DEM, DTM, and DSM models ($r^2 < 0.8$). However, limited information on the application of multispectral-based plant height estimation using DEM models suggests comparable levels of accuracy ($r^2 = 0.90$–0.96) with those estimated from RGB-based 3D point cloud models.

*5.2. Canopy Cover and Leaf Area Index*

Canopy cover and LAI are associated with crop nutritional status and are closely related to crop yield [25] because rapid canopy coverage and LAI improve light interception [177], conserve soil moisture by reducing soil evaporation [178], and suppress weeds [179]. Similar to plant height estimation, canopy cover and LAI measurements from sUAS platforms were often conducted using RGB and multispectral sensors, while LiDAR sensors were often used with terrestrial platforms. When using sUAS platforms, canopy cover and LAI prediction accuracy (quantified using $r^2$ values) using RGB sensors ranged from 0.70–0.83 for blueberry [121], soybean [69], and maize [122]. Ampatzidis and Partel [87] obtained a prediction accuracy of 85% for canopy cover using multispectral imagery with an artificial neural network in citrus. Fenghua et al. [70] reported an $r^2$ value of 0.82 and an RMSE of 10% for estimating LAI using hyperspectral imagery in rice; however, a reduced error (RMSE 5.9%) was reported for LAI using RGB and multispectral data fusion in soybean [50]. LAI and canopy cover estimation accuracy were consistently higher with terrestrial sensing than with sUAS sensing regardless of the imaging sensor deployed. The reported accuracies with terrestrial-based sensing for canopy cover or LAI were reported in several crops. Examples include cotton (LiDAR, $r^2 = 0.97$) [154], wheat (LiDAR, $r^2 = 0.92$) [158], soybean (RGB, $r^2 = 0.89$), (Hue, Saturation, and Intensity, $r^2 = 0.80$) [161], maize (LiDAR, $r^2 = 0.92$) [162], and sorghum (LiDAR, $r^2 = 0.94$) [162].

Unlike plant height estimation, the prediction accuracy for plant canopy cover and LAI tended to be higher and more consistent with terrestrial platforms compared to sUAS. This is because captured images nearer the canopy tend to have higher resolution (especially along the edges or boundaries of the canopy or leaf surface) compared to those captured at higher altitudes. This suggests that the distance between the sensor and the crop canopy is likely an important factor for both LAI and canopy cover estimation in agricultural crops. Both image analysis and machine-learning algorithms showed promise for canopy cover and LAI estimation under field conditions. Based on the information gathered from several studies using both platforms, LiDAR, RGB, and multispectral imaging sensors play important roles in both canopy crop cover and LAI estimation. Thus, given the sensor's simplicity and the ease of adaptation to sUAS platforms, both RGB and multispectral sensors are suggested for both crop LAI and canopy cover estimation. Xie and Yang [45] reached a similar conclusion in their review on sensor performance for LAI estimation using an sUAS platform.

*5.3. Biomass*

Plant biomass is the amount of plant dry matter accumulated per unit area and is a critical indicator of plant growth, radiation use efficiency, and productivity. The reviewed HTPP studies (which used both sUAS and terrestrial platforms) on plant biomass or dry matter production were conducted on different crop types, including grain (maize, rice, wheat, barley, soybean, and triticale), root or tuber (cassava and sugar beet), legume

(dry bean), and an energy crop (*Arundo donax*) (Table 3). In an sUAS-based multi-sensor (RGB, multispectral and thermal) data fusion study on biophysical variables in soybean, Maimaitijiang et al. [50] reported that multispectral and thermal data fusion had the best performance (RMSE 10.2%) in the estimation of soybean biomass production. The study also reported that data fusion using extreme machine learning-based regression (ELR) for the estimation of plant biomass traits improved prediction accuracy compared to partial least squares regression (PLSR) and support vector regression (SVR) models. A similar level of accuracy (RMSE 11%) was reported using an sUAS-based hyperspectral sensor in rice [70]. Wang et al. [108] reported that the accuracy of biomass estimation in maize was much higher using an sUAS LiDAR sensor ($r^2 = 0.83$) compared to hyperspectral ($r^2 = 0.47$); however, data fusion from both sensors improved prediction accuracy ($r^2 = 0.88$) compared to the individual sensor. Data fusion from sUAS-based multispectral and thermal sensors resulted in a strong negative correlation ($r^2$ ranged from $-0.67$ to $-0.91$) with GNDVI in dry bean [51], while an $r^2$ value of 0.71 was reported for biomass estimation in a bioenergy crop (*Arundo donax*) using an airborne LiDAR sensor. Prediction accuracy for plant biomass production under the terrestrial platforms also varied among sensors and crop types. Prediction accuracy as determined by $r^2$ values was 0.92–0.93 using LiDAR sensors in bread wheat [158], 0.73 in cassava [164], and 0.68–0.80 in maize [165], while $r^2$ values were from 0.82–0.88 for RGB imagery in sugar beet [163].

Regardless of sensor types, results from these studies suggest that plant biomass can be estimated with a high level of accuracy from either sUAS or terrestrial platforms. Although data captured using single RGB and LiDAR sensors performed well in estimating plant biomass [163,180], integrating data fusion from multiple sensors (thermal with multispectral or hyperspectral) can further improve prediction accuracies [50,51]. Therefore, future studies with a focus on exploring multiple sensor fusion could be a promising strategy for improving the accuracy of plant biomass estimation in HTPP research.

*5.4. Yield Estimation*

Crop yield is an important trait for phenotyping as it is closely related to the development and differentiation of organs as well as the distribution and accumulation of photosynthetic products [15]. Thus, it is a major trait for farmers and a core focus in crop science research [15]. Crop yield prediction models can be established by combining plant physiological parameters with vegetation indices. Commonly adopted parameters include crop height, biomass, LAI, chlorophyll content, and length of growing period [25]. Du and Noguchi [166] used sUAS-based multi-temporal color imaging to monitor wheat growth. The authors used this information to develop an in-field spatial variation map of wheat yield. The result showed that wheat yield was highly correlated ($r^2 = 0.94$, RMSE 0.02) with four vegetation indices: visible-band difference vegetation index (VDVI), normalized green-blue difference index (NGBDI), green-red ratio index (GRRI), and excess green vegetation index (ExG). Zhou et al. [167] also reported that a multi-temporal linear regression model improved rice yield prediction using two vegetation indices (Visible Atmospherically Resistant Index (VARI) for RGB and NDVI for multispectral sensors). The authors also indicated that rice yield prediction accuracy was higher using a multispectral sensor ($r^2 = 0.76$) compared to RGB ($r^2 = 0.73$). Maresma et al. [181] developed a linear regression model between vegetation indices (green-band- and red-band-based indices derived from different sUAS-based multispectral sensors) and crop height to predict grain yield in maize. The authors reported a high and significant prediction accuracy ($r^2 = 0.82$, RMSE = 0.15, $p < 0.001$). Hu et al. [144] compared the performance of an sUAS with a multispectral sensor and a terrestrial platform with a hyperspectral sensor for predicting grain yield in wheat using NIR-based spectral indices. The authors reported that data collected from the sUAS showed consistently higher levels of correlation with grain yield ($r^2 = 0.60$, 0.63, and 0.65 for NDVI, NIR:Red, and NIR:Green, respectively) compared to those derived from the terrestrial platform ($r^2 = 0.54$, 0.52, and 0.52 for NDVI, NIR:Red, and NIR:Green, respectively).

In recent years, multispectral imaging has become a common way of extracting crop traits such as crop water and nutrient contents, LAI, and yield [25]. Hence, most studies (retrieved in this review) that estimated crop yield from both sUAS and terrestrial platforms were conducted using multispectral sensors. Compared to RGB sensors, multispectral sensors provide more useful wavebands in both visible and NIR spectral regions and, therefore, can be more useful in crop yield estimation in HTPP research [45]. A few studies on crop yield estimation were also conducted using hyperspectral sensors [135,182].

Crop yield estimation with an sUAS platform was found to be more accurate compared to a terrestrial imaging platform [144]. Higher yield prediction accuracy for the sUAS platform could be due to one or more of the following factors. First, the removal of non-vegetation pixels (especially from images with fewer plant rows) could be better achieved from sUAS imagery compared to the images from the terrestrial system. This could be achieved via a number of established classification algorithms such as Support Vector Machine (SVM) classification or CNNs [183,184]. Second, sUAS sensing had an advantage over terrestrial platforms in generating real-time surface maps compared to a more time-consuming approach for the terrestrial platform, especially for large field areas (>2 ha) [168,185]. Third, using high-resolution sensors and low altitudes, sUAS can overcome the major limitations of terrestrial platforms. This includes non-simultaneous measurement of multiple plots, vibration from rough field surfaces, and challenges associated with maneuverability in the field [168,186].

## 6. Suggested Directions for Future Research

Given the status and challenges of sUAS platforms for HTPP, the following future considerations are hereby discussed. First, technological improvements in both the operations of sUAS and onboard sensors are essential. The availability of technologies that allow for higher flight stability, longer flight duration (more durable power source), and greater payload capacity is needed for sUAS-based phenotyping over large fields. Additionally, the availability of low-cost and high-performance sensors is critical. The cost of imaging sensors varies with type and complexity; hence, researchers often include sensor protection features (in case of a crash) onboard the sUAS. These additional features may increase payload and limit the number of onboard sensors; hence, sensor fusion (which can improve the accuracy of sUAS phenotyping) on a single flight may be more challenging.

Second, a significant proportion of the previous sUAS-based phenotyping studies have evaluated chlorophyll content as an important pigment for diagnosing the state of photosynthesis activity and indicating the general health of the plant. However, no study has been conducted on the application of sUAS-based imagery for the evaluation of carotenoids, which play a significant role in plant photosynthetic processes, stress signals, and adaptation, as well as for crop defense mechanisms [187–189]. Therefore, sUAS-based studies addressing the potential effects of carotenoids on stress resilience and crop protection may provide additional benefits of HTPP in plant breeding.

Third, the complexity of sUAS-based data processing and analysis may limit its use among researchers because no commercially available fully automated data processing or analysis pipelines are known. Therefore, studies with a focus on the development of fully automated pipelines of sUAS-based crop imagery will enhance the impacts and applications of sUAS technology for phenotyping, crop monitoring, and improving crop management practices. Another area of potential consideration is the application of sUAS for HTPP in fruit and vegetable crops. Compared to row or grain crops, a small proportion of the sUAS-based phenotyping studies were conducted on horticultural (fruit and vegetable) crops. Limited application of HTPP in horticultural crops could be due to the unique growing conditions (controlled environment, netting) and/or short production cycles often associated with these crops. Given the high dependencies of these crops on pest and disease control practices, a wider application of sUAS systems may enhance the early detection of diseases and potentially reduce the frequency of pesticide applications, thereby reducing the chemical load on the environment.

## 7. Discussion

High-throughput plant phenotyping (HTPP) has received increased attention from research programs to determine and evaluate important crop phenotypes associated with growth and productivity under different growing conditions (limited water and nutrients, heat stress, salinity stress, disease pressure, etc.). Research programs considering HTPP adoption will need to carefully consider the best phenotypes to capture for their crops and objectives, the most efficient sensors, and platforms to capture those phenotypes, the physical size and location(s) of their field trials, the necessary human and computational resources associated with each sensor/platform combination, and the costs for equipment procurement and maintenance. A summary of these considerations for the two platforms (sUAS and terrestrial) and sensors evaluated in the literature is presented in Table 4. The evaluation criteria for the platforms included initial costs and those associated with maintenance, the degree of training required by operators, the relative number of personnel needed (Human Resources), the payload capabilities, and the general coverage area. The evaluation criteria for the sensors also included costs, a difficulty rating associated with the deployment of the sensors, the platforms the sensors are compatible with, and the data processing and computational resources needed for phenotype extraction.

**Table 4.** Summary of platform and sensor considerations gathered from the reviewed journal articles.

| Platform | Initial Cost | Maintenance | Training | Human Resources | Payload | Coverage Area |
|---|---|---|---|---|---|---|
| sUAS | Moderate (≤USD 20K) | Low to Moderate | Moderate to High | Low | Low (≤1 kg) | Moderate to High (>5 ha) |
| Terrestrial handheld | Low to Moderate (USD 100–20K) | Low | Low | High | Low to Moderate (1–20 kg) | Low (<2 ha) |
| Terrestrial cart | Moderate (≤USD 20K) | Low to Moderate | Moderate | Low to Moderate | Moderate (≤20 kg) | Low to Moderate (≤2 ha) |
| Terrestrial tractor | High (≥USD 40K) | High | Moderate to High | Moderate | High (>20 kg) | Moderate (≤5 ha) |
| **Sensor** | **Initial Cost** | **Maintenance** | **Deployment** | **Platform** | **Data Processing** | **Computational Resources** |
| RGB camera | Low (≤USD 1K) | Low | Easy | sUAS * and Terrestrial | Easy to Moderate | Moderate |
| Multispectral camera | Moderate (≤USD 20K) | Moderate | Moderate | sUAS * and Terrestrial | Moderate | Moderate |
| Thermal camera | Moderate (≤USD 20K) | Moderate | Moderate to Difficult | sUAS * and Terrestrial | Moderate | Moderate to High |
| Hyperspectral camera | High (≥USD 40K) | Moderate | Moderate to Difficult | sUAS | Moderate | Moderate to High |
| LiDAR | High (≥USD 40K) | Moderate to High | Difficult | sUAS and Terrestrial * | Moderate to Difficult | Moderate to High |
| Infrared thermometer | Low (≤USD 1K) | Low | Easy | Terrestrial | Easy | Low |
| Multispectral radiometer | Low (≤USD 1K) | Low | Easy | Terrestrial | Easy | Low |
| Ultrasonic transducer | Low (≤USD 1K) | Low | Easy | Terrestrial | Easy | Low |
| Laser scanner | Low to Moderate (USD 100–20K) | Low to Moderate | Moderate | Terrestrial | Easy to Moderate | Low to Moderate |

* Indicates the more commonly used platform.

The ratings for each category were determined using several different criteria for each category. The costs, payload capabilities, general coverage area, and sensor platform compatibility were all gathered from the reviewed literature and/or visiting vendor and research group websites. The human resources category was primarily rated by the author's personal experience with the various platforms and includes consideration for the number of personnel to operate the equipment at a given time, as well as maintenance of the platform. For example, the handheld terrestrial rating is 'high' because multiple personnel (at least five) was considered necessary to achieve a relatively 'high-throughput' capture rate, while the 'moderate' rating for the terrestrial tractor was given because two to three

people with mechanical expertise may be needed to maintain the equipment as well as a trained operator for data collection. The ratings for the remaining sensor categories (deployment, data processing, and computational resources) were interpreted from the materials and methods sections of the reviewed literature and somewhat from the authors' personal experience. The evaluation criteria provided in Table 4 are meant to be used as a 'quick guide' to help future HTPP adopters identify a 'good fit' for the current technology for their program given their phenotypes of interest.

The most common phenotypes evaluated with HTPP found in the literature were growth traits (canopy height, cover, LAI, and biomass) which could be estimated from both sUAS and terrestrial platforms with a high level of accuracy from either a single sensor or a complex of sensors (sensor fusion). Of the phenotypes evaluated, canopy height received the most attention from studies across multiple crops. While height is an important indicator of crop health and yield potential, height by itself may provide limited information to research programs; therefore, sensors that only provide canopy height information might not be as useful in terms of labor and cost efficiency. Pairing single phenotype sensors with other sensors, e.g., a laser scanner and thermal camera, can provide more information for the field work expended but adds complexity, which may prevent some programs from utilizing this approach if they do not have the appropriate human resources (e.g., mechanical and/or electrical engineering expertise).

HTPP studies that relied on imaging sensors, such as RGB, hyperspectral and multispectral cameras, and LiDAR, were able to estimate multiple phenotypes at the same time with varying degrees of accuracy. Compared to other sensors, RGB and multispectral cameras were used most often on both sUAS and terrestrial platforms, possibly due to their lower cost, operational simplicity, and relatively easy data collection and analysis (Table 4). However, since data from RGB sensors are limited to only three wavebands (red, green, and blue), their application may be limited for some traits. Thus, for the collection of other spectral bands, the application of multispectral sensors for HTPP is receiving more attention among researchers. Additionally, the application of hyperspectral sensors is receiving increased attention due to the abundance of spectral information (from hundreds or even thousands of wavebands), hence providing more spectral detail than RGB and multispectral imaging. However, the application of hyperspectral sensors for HTPP may be limited due to cost and associated data management complexities (Table 4). Overall imaging sensors require increased expertise for data processing and management than simple sensors and may require increased computational resources (e.g., high-performance computing (HPC)). Therefore, programs with limited access to these types of resources may currently be prevented from adopting this technology.

## 8. Conclusions

Over the last decade, interest in and adoption of HTPP has increased, which is, in part, due to the relatively inexpensive and easy deployment of sUAS with onboard RGB cameras. These 'out of the box' systems are less expensive and generally require less expertise than the custom-built multi-sensor terrestrial tractors that were initially developed for HTPP. However, like the tractors and less expensive carts that followed, sUAS have limitations (payload and wind) that may prevent some programs from adopting this technology. Given the rapid advancement in sUAS technology and available sensors, more programs will likely be able to incorporate sUAS in the future. Advancements in artificial intelligence and machine learning will likely increase the adoption rate of HTPP as more complex phenotypes can be extracted from image analysis. The next (ongoing) challenge for HTPP will be data storage, management, and processing to fully address the gap between phenotype and genotype data.

**Author Contributions:** Conceptualization, I.T.A., K.R.T. and A.L.T.; methodology, I.T.A., K.R.T. and A.L.T.; formal analysis, I.T.A.; investigation, I.T.A.; data curation, I.T.A.; writing—original draft preparation, I.T.A.; writing—review and editing, I.T.A., K.R.T. and A.L.T.; supervision, K.R.T. and A.L.T. All authors have read and agreed to the published version of the manuscript.

**Funding:** This research was funded by the United States Department of Agriculture-Agricultural Research Service project numbers 2020-2100-013-00D and 2020-13660-008-00D.

**Data Availability Statement:** Not applicable.

**Acknowledgments:** This research was supported in part by an appointment to the Agricultural Research Service (ARS) Research Participation Program funded by ARS SCINet and administered by the Oak Ridge Institute for Science and Education (ORISE) through an interagency agreement between the U.S. Department of Energy (DOE) and the U.S. Department of Agriculture (USDA). ORISE is managed by ORAU under DOE contract number DE-SC0014664. All opinions expressed in this paper are the author's and do not necessarily reflect the policies and views of USDA, DOE, or ORAU/ORISE.

**Conflicts of Interest:** The authors declare no conflict of interest. Mention of trade names or commercial products in this publication is solely for the purpose of providing specific information and does not imply recommendation or endorsement by the USDA. The USDA is an equal opportunity employer.

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
