# Peer review of "Advances in the Application of Small Unoccupied Aircraft Systems (sUAS) for High-Throughput Plant Phenotyping"

_remotesensing, doi:10.3390/rs15102623_

Round 1
Reviewer 1 Report
The authors have made an intense review on the application of drone technologies used in precision farming to identify the advantages and limitations of terrestrial platforms. The paper has a well-designed research and technical infrastructure. But in order to have a final decision, the author(s) should do the following minor revisions:
v At the end of the introduction, authors have to highlight the contribution of their paper.
v Review papers are very demanding papers to write and they should propose a synthesis of the applications provided and not stating the description of the collection of applications. There are numerous review papers for every single chapter of your review; this review is too broad.
v The first letter of the expansions of the abbreviations should be capitalized in line 66 & 90.
v A short paragraph detailing the structure of the paper can be added at the end of the introduction to help the readers to localize information along the paper.
v The information about the methodology followed to search the included papers must be added. Authors should precise the used keywords, the used search tools, if they limit their search to a certain period of time, etc…This can be included in sections 1 or 2.
v More figures and comments have to be added. Authors can analyze which are the most studied crops, the most used AI technique, the most common picture size, the most common picture acquisition type, etc…Thus, authors can provide a general view of the topic.
v I suggest creating more tables in which the authors summarize the used technique for different types of fruit.
v Please add reference study using machine learning techniques and deep learning.
v Some of the intermediate results can go in supplementary document or removed to enhance the document readability.
v Discuss the major results. Capture some limitations spanning the intermediate results.
v The paper should indicate how the current work can be scaled up or can prove to be utilitarian for other kinds of work. The authors can suggest limitations while indicating the same.
Author Response
Reviewer 1
Comments and Suggestions for Authors
The authors have made an intense review on the application of drone technologies used in precision farming to identify the advantages and limitations of terrestrial platforms. The paper has a well-designed research and technical infrastructure. But in order to have a final decision, the author(s) should do the following minor revisions:
v At the end of the introduction, authors have to highlight the contribution of their paper.
The information about the contribution of this paper as well as how our review differs from the previous review papers are detailed in the submitted manuscript. This information was provided at the end of the introduction section as suggested in the comments. Please see lines 122 -149.
v Review papers are very demanding papers to write and they should propose a synthesis of the applications provided and not stating the description of the collection of applications. There are numerous review papers for every single chapter of your review; this review is too broad.
As described in first section of the paper, we used a specific methodology to retrieve related papers in the literature. We agree review papers should synthesize the literature, and the aim of our study.
v The first letter of the expansions of the abbreviations should be capitalized in line 66 & 90.
Thank you for the detailed review of this manuscript. The abbreviations were corrected as suggested. Now on lines 67 &91.
v A short paragraph detailing the structure of the paper can be added at the end of the introduction to help the readers to localize information along the paper.
We thank you for this suggestion. This information was added to the revised manuscript. Please see lines 123 – 129.
v The information about the methodology followed to search the included papers must be added. Authors should precise the used keywords, the used search tools, if they limit their search to a certain period of time, etc…This can be included in sections 1 or 2.
The authors agree with your suggestion. A detailed methodology followed during the search for materials used in this manuscript including the search engine tools, keywords, the timeframe of the search, total number of articles retrieved during the timeframe etc. were detailed in the submitted manuscript. Please see this information from lines 150-163.
v More figures and comments have to be added. Authors can analyze which are the most studied crops, the most used AI technique, the most common picture size, the most common picture acquisition type, etc…Thus, authors can provide a general view of the topic.
We ask the reviewer to please provide clarification. The second comment above by the reviewer mentions that the review is too broad. This suggestion seems to request inclusion of more information. Which is preferred?
v I suggest creating more tables in which the authors summarize the used technique for different types of fruit.
The information about the HTPP techniques, including imaging, processing, and their accuracies were described by crop, platform, and senor in tables 2 and 3 of the manuscript.
v Please add reference study using machine learning techniques and deep learning.
Articles that focused on the application of machine learning and deep learning as related to HTPP were referenced in the submitted manuscript. Examples of these references include articles 88, 118, 139, 187.
v Some of the intermediate results can go in supplementary document or removed to enhance the document readability.
Please provide clarification on ‘intermediate results’. Is the reviewer referring to Table 1, 2, or 3?
v Discuss the major results. Capture some limitations spanning the intermediate results.
Please provide clarification on ‘intermediate results’. General discussions are provided in each sub-section from 2-5. A broader discussion including limitations is provided in sections 6 and 7.
v The paper should indicate how the current work can be scaled up or can prove to be utilitarian for other kinds of work. The authors can suggest limitations while indicating the same.
Suggestions for future work and limitations are provided in sections 6 and 7.
We would like to thank the reviewer for their time and attention.
Reviewer 2 Report
The authors of the manuscript titled "Advances in the application of small unoccupied aircraft systems (sUAS) for high-throughput plant phenotyping" have prepared a well-structured and well-written review of published research. There are some minor typo's (e.g. Line 306) , and I would like to raise a few minor issues, to improve the manuscript:
1. provide a definition of sUAS and how they differ from UAVs.
2. Lines 237-238: I would specify that you mean a peak in reflectance, just to avoid any confusion.
3. Lines 244-246: While this statement isn't incorrect, it does omit the far more influental atmospheric water vapour absorption. Also, in regard to the spectral signatures of plants in the SWIR region; this part of the spectrum carries information about biochemical properties of plants, and is crucial for any chemometric analysis. Water absorption limits this aspect, but doesn't prevent it completely. I would recommend to rephrase this sentence and add more information to it, as discussed abouve.
4. In Section 2 you talk about LiDAR for 3D structures, but you don't mention point-clouds derived from RGB or multispectral sensors. You do mention them later in the manuscript (around Line 363). I would recommend to include them here, in Section 2.
5. Section 6: Research is focused on economically and nutritionally more important crops. Maize, wheat, soy, rice, and potatoes provide most of the calories to feed humanity, and require large areas for production. Fruit and vegetable crops are not that important in this regards, and some vegetables can be grown very succesfully in controled environments, e.g. tomatoes. Another aspect to consider are hail nets, which often cover these plants, and limit the use of remote sensing methods.
5.1 Another thing to consider is the availability of data. Open science is the way forward, but the data from most published research isn't available or has restricted access. Even though there are a lot of published results available, most of them with very good accuracies, how much of that is actually in use? Or rather, has a TRL of more than 5 or 6? How much of that research (data, models, code) is publicly available, following FAIR principles?
6. Discussion: The first two paragraphs of the discussion are basically a description of the methodology for Table 4. The last two are a summary of the manuscript. Since some things have been discussed in the previous Sections of the manuscript, I would recommend to remove the discussion as a separate section, and incorporate it in the previous parts.
6.1 I would expect a more thorough review and discussion of the generalizability of any of these methods. Is there any method that has been verified on several varieties and works acceptably accurately on all of them? Any methods that were verified on datasets from different years and geographical regions? This is where data sharing and cooperation are crucial and the only way to more forward.
7. Table 4: Please provide actual values, at least for costs (as you did for payloads). Also, for Low to Moderate payloads you wrote "1 > 20". The way it's written it would mean that 1 is more than 20. I understand what you meant (between 1 and 20), but please correct this.
Author Response
Reviewer 2
Comments and Suggestions for Authors
The authors of the manuscript titled "Advances in the application of small unoccupied aircraft systems (sUAS) for high-throughput plant phenotyping" have prepared a well-structured and well-written review of published research. There are some minor typo's (e.g. Line 306) , and I would like to raise a few minor issues, to improve the manuscript:
1. provide a definition of sUAS and how they differ from UAVs.
Small unmanned aircraft systems (sUAS) is the name and acronym used by the Federal Aviation Administration (FAA) in the U.S. A small unmanned aircraft is an unmanned aircraft weighing less than 55 pounds on takeoff, including everything that is on board or otherwise attached to the aircraft. A small unmanned aircraft system (small UAS) means a small unmanned aircraft and its associated elements (including communication links and the components that control the small unmanned aircraft) that are required for the safe and efficient operation of the small unmanned aircraft in the national airspace system. The term Unmanned Aerial Vehicles (UAV) was presented by the U.S. military in the 1950s and is a very broad definition for anything that flies without a physical pilot in the vehicle. At some point the DoD adopted the term Unmanned Aircraft Systems and began classifying the systems from small to large based on weight. Both UAV and UAS terms are technically correct, however, UAV seems to be falling out of favor due to FAA regulations. The sUAS term is more descriptive for the platforms used in HTPP research.
2. Lines 237-238: I would specify that you mean a peak in reflectance, just to avoid any confusion.
Thank you for this suggestion, we have changed the wording to …demonstrates a reflectance peak in the visible green region… on line 240.
3. Lines 244-246: While this statement isn't incorrect, it does omit the far more influental atmospheric water vapour absorption. Also, in regard to the spectral signatures of plants in the SWIR region; this part of the spectrum carries information about biochemical properties of plants, and is crucial for any chemometric analysis. Water absorption limits this aspect, but doesn't prevent it completely. I would recommend to rephrase this sentence and add more information to it, as discussed abouve.
Thank you for this suggestion. We have added the following sentence in lines 249-252.
However, when relying on solar illumination, absorption of radiation by atmospheric water vapor at 1380 nm and 1870 nm can drastically reduce signal at these wavelengths, which can impede the analysis of the water absorption features of vegetation spectra.
4. In Section 2 you talk about LiDAR for 3D structures, but you don't mention point-clouds derived from RGB or multispectral sensors. You do mention them later in the manuscript (around Line 363). I would recommend to include them here, in Section 2.
Section 3 provides a brief overview of data analysis and covers generation of point clouds (Lines 311-320). Section 4.1 covers traits have been extracted by using 3-D (point cloud) analysis (368-375).
5. Section 6: Research is focused on economically and nutritionally more important crops. Maize, wheat, soy, rice, and potatoes provide most of the calories to feed humanity, and require large areas for production. Fruit and vegetable crops are not that important in this regards, and some vegetables can be grown very succesfully in controled environments, e.g. tomatoes. Another aspect to consider are hail nets, which often cover these plants, and limit the use of remote sensing methods.
Thank you for this important observation. This review covers HTPP for all crops reported (including vegetable crops) in the literatures. Although vegetable crops may not be considered as high in calories compared to grain and tuber crops as mentioned in the comments, Efforts towards improving these crops have proven to have major economic and environmental impacts on a global scale. We recognize that the application of sUAS-based HTPP may be limited under certain growing conditions that are unique to vegetable crops (controlled environment, netting etc.), however, on a global scale, a vast majority of vegetable crops are produced under open-field conditions where HTPP technology could be easily applied. Hence, vegetable crops are considered in the manuscript. The potential limitations of sUAS-based HTPP for vegetable crops are added to the revised manuscript. Please see lines 736-738.
5.1 Another thing to consider is the availability of data. Open science is the way forward, but the data from most published research isn't available or has restricted access. Even though there are a lot of published results available, most of them with very good accuracies, how much of that is actually in use? Or rather, has a TRL of more than 5 or 6? How much of that research (data, models, code) is publicly available, following FAIR principles?
Again, thank you for this very important point. Unfortunately, the input data from most of the published articles tends not to be publicly or readily available as indicated in the comment above. Hence, possible limitation to the impacts or applications of these technologies on a global scale. Over the years, the understanding of this challenge has resulted in the creation of several information sharing platforms where research data, models, codes etc.) can be shared among researchers. Another reason why this information isn’t available in the public domain is because of the patent on developed programs for commercial purposes. These potential challenges have been addressed in the manuscript please see lines 728-733.
6. Discussion: The first two paragraphs of the discussion are basically a description of the methodology for Table 4. The last two are a summary of the manuscript. Since some things have been discussed in the previous Sections of the manuscript, I would recommend to remove the discussion as a separate section, and incorporate it in the previous parts.
We respectfully disagree. We prefer the first two paragraphs remain in the discussion as they elaborate more broadly from the specific results presented in the results section.
6.1 I would expect a more thorough review and discussion of the generalizability of any of these methods. Is there any method that has been verified on several varieties and works acceptably accurately on all of them? Any methods that were verified on datasets from different years and geographical regions? This is where data sharing and cooperation are crucial and the only way to more forward.
We agree with the reviewer, however no such processing pipelines, datasets, or data sharing initiatives were found in the literature reviewed. As above, these challenges were added to the manuscript, please see lines 728-733.
7. Table 4: Please provide actual values, at least for costs (as you did for payloads). Also, for Low to Moderate payloads you wrote "1 > 20". The way it's written it would mean that 1 is more than 20. I understand what you meant (between 1 and 20), but please correct this.
Cost ranges were added. Some minor adjustments were made to accommodate a consistent price range for both platforms and sensors.
Thank you for bringing this to our attention, we have replaced 1 > 20 with a 1 – 20.
We would like to thank the reviewer for their time and attention!
Reviewer 3 Report
1 Line 9: “Correspondence” written 2 times, remove one.
2 Line 144, is it correct?
3 Lines 123-127, is there any duplication with the abstract? Same sentence
4 Line 173, Table1 statistics should have units.
5 Is the full text level 1 heading written appropriately? Please revise according to the template.
6 Line 294-339, suggest drawing a graph or table to be more visual and effective.
7 Line 341, space problem.
8 Line 383, Is there a problem with Table 2 being placed here?
9 Line 416, is improving water productivity correct?
10 For abbreviations, just write the full name the first time and use the abbreviation the second time.
11 Revise the references according to the template format.
12 Add an image to enhance the readability and interest of the article.
Author Response
Reviewer 3
Comments and Suggestions for Authors
1 Line 9: “Correspondence” written 2 times, remove one.
Thank you for bringing this to our attention, it has been corrected.
2 Line 144, is it correct?
Yes, this information in the referenced line is correct as the manuscript provides a comprehensive review of more recent studies in HTPP research within the search timeframe. Many of the retrieved articles are cited and included in the reference lists.
3 Lines 123-127, is there any duplication with the abstract? Same sentence
Thank you for a detailed review of this manuscript. The abstract is not duplicated in any part of the manuscript. Included in the abstract are the objectives of this review, and this information is reiterated at the end of the introduction to ensure a good readability and flow of the manuscript.
4 Line 173, Table1 statistics should have units.
The required unit (percentage) was included in the referenced table. Please see the column headings.
5 Is the full text level 1 heading written appropriately? Please revise according to the template.
Thank you, the formatting was revised according to the template.
6 Line 294-339, suggest drawing a graph or table to be more visual and effective.
The description provided is a general overview for image data processing and not a specific procedure that is agreed upon within the sUAS community. A flowchart seems to make it overly specific for a review paper.
7 Line 341, space problem.
The indent problem was corrected.
8 Line 383, Is there a problem with Table 2 being placed here?
The duplicated caption for table 2 placed in the text was removed.
9 Line 416, is improving water productivity correct?
Yes, defined as the physical or economic output per unit of water application.
10 For abbreviations, just write the full name the first time and use the abbreviation the second time.
We have reviewed the manuscript several times and are not seeing repeated use of the full name after the abbreviation was introduced. There are some occasions where the abbreviation is re-introduced if it had not been used for several sections. If the reviewer finds situations where the abbreviation should have been used please provide the line number and we will make the correction.
11 Revise the references according to the template format.
The references were corrected to the journal format.
12 Add an image to enhance the readability and interest of the article.
We thank you for this suggestion, could you please be more specific? What would you like an image of?
We would like to thank the reviewer for their time and attention!
Round 2
Reviewer 3 Report
The author responded well to the questions raised and achieved the effect of prediction. This article is available for acceptance.